# Description of a Newly Isolated *Blautia* *faecis* Strain and Its Benefit in Mouse Models of Post-Influenza Secondary Enteric and Pulmonary Infections

**DOI:** 10.3390/nu14071478

**Published:** 2022-04-01

**Authors:** Sophie Verstraeten, Valentin Sencio, Audrey Raise, Eugénie Huillet, Séverine Layec, Lucie Deruyter, Séverine Heumel, Sandrine Auger, Véronique Robert, Philippe Langella, Laurent Beney, François Trottein, Muriel Thomas

**Affiliations:** 1Micalis Institute, Institut National de Recherche pour L’agriculture, L’alimentation et L’environnement (INRAE), AgroParisTech, Université Paris-Saclay, UMR1319, F-78350 Jouy-en-Josas, France; sophie.verstraeten@inrae.fr (S.V.); eugenie.huillet@inrae.fr (E.H.); severine.layec@inrae.fr (S.L.); sandrine.auger@inrae.fr (S.A.); veronique.robert@inrae.fr (V.R.); philippe.langella@inrae.fr (P.L.); 2Paris Center for Microbiome Medicine (PaCeMM) FHU, AP-HP, F-75571 Paris, France; 3U1019-UMR 9017-CIIL-Center for Infection and Immunity of Lille, University Lille, F-59000 Lille, France; valentin.sencio@gmail.com (V.S.); lucie.deruyter@pasteur-lille.fr (L.D.); severine.heumel@pasteur-lille.fr (S.H.); francois.trottein@pasteur-lille.fr (F.T.); 4Centre National de la Recherche Scientifique (CNRS), UMR9017, F-59000 Lille, France; 5Institut National de la Santé et de la Recherche Médicale (Inserm) U1019, F-59000 Lille, France; 6Centre Hospitalier Universitaire de Lille, F-59000 Lille, France; 7Institut Pasteur de Lille, F-59000 Lille, France; 8Procédés Alimentaires et Microbiologiques, Institut Agro, University Bourgogne Franche-Comté, UMR A 02.102, F-21000 Dijon, France; audrey.raise@u-bourgogne.fr (A.R.); laurent.beney@u-bourgogne.fr (L.B.)

**Keywords:** *Blautia faecis*, acetate, post-influenza, pulmonary and enteric superinfection, anti-inflammatory, extremely oxygen-sensitive probiotics

## Abstract

The expanding knowledge on the systemic influence of the human microbiome suggests that fecal samples are underexploited sources of new beneficial strains for extra-intestinal health. We have recently shown that acetate, a main circulating microbiota-derived molecule, reduces the deleterious effects of pulmonary *Streptococcus pneumoniae* and enteric *Salmonella enterica* serovar Typhimurium bacterial post-influenza superinfections. Considering the beneficial and broad effects of acetate, we intended to isolate a commensal strain, producing acetate and potentially exploitable in the context of respiratory infections. We designed successive steps to select intestinal commensals that are extremely oxygen-sensitive, cultivable after a freezing process, without a proinflammatory effect on IL-8 induction, and producing acetate. We have identified the *Blautia faecis* DSM33383 strain, which decreased the TNFα-induced production of IL-8 by the intestinal epithelial cell line HT-29. The beneficial effect of this bacterial strain was further studied in two preclinical models of post-influenza *Streptococcus pneumoniae* (*S.p*) and *Salmonella enterica* serovar Typhimurium (*S.t*) superinfection. The intragastrical administration of *Blautia faecis* DSM33383 led to protection in influenza-infected mice suffering from an *S.p*. and, to a lesser extent, from an *S.t* secondary infection. Altogether, this study showed that *Blautia faecis* DSM33383 could be a promising candidate for preventive management of respiratory infectious diseases.

## 1. Introduction

It is estimated that around 10^12^ bacterial species exist in the intestinal microbial environment, of which approximately 50% of bacteria are easily cultivated, and even less are studied in detail [1]. Intestinal bacteria are difficult to cultivate mainly because of their strict anaerobic nature and also their nutritional needs. Meanwhile, it is important to overcome the difficulty in isolating, culturing, and producing intestinal species with health benefits to generate so-called next-generation probiotics [2]. *Faecalibacterium prausnitzii* and *Dysosmobacter welbionis* are both known to be abundant in healthy people but reduced in diseased individuals [3,4]. *F. prausnitzii* is of interest for its anti-inflammatory properties [5,6,7], and *D. welbionis* for its protective effect against metabolic disorders [3]. Our rapidly expanding knowledge on the human microbiome suggests that a large panel of beneficial anaerobe strains may be isolated from the dominant members of our adult microbiota [8]. With this in mind, we sought to isolate new intestinal bacterial strains from human fecal samples.

Short chain fatty acids (SCFA) are mainly produced by the microbiota and can be found in different ratios in the healthy gut, depending on the composition of the microbiota and the diet of an individual. The SCFA most present in the gut is acetate, with propionate and butyrate varying to have the second largest concentration present [9]. Once produced in the gut, these SCFAs are used by intestinal epithelial cells and the gut microbes and are exported via the portal vein. In humans, it has been estimated that the portal vein fluxes reflect SCFA production in the gut, but with a lower concentration of butyrate after its uptake for metabolism in the colonic mucosa [10]. After further passage of the blood through the liver, propionate is taken up as well, whereas acetate is less metabolized by the liver and consequently is found in significant amounts in systemic circulation as an energy substrate for the rest of the body [11,12]. Acetate is a main circulating microbiota-derived molecule that influences extra-intestinal tissues [13], implying that acetate-producing bacteria have systemic health effects for the host.

Acetate is an important energy substrate for both human cells and bacteria in the gut and consequently a modulator of host metabolism and microbial ecology. Acetate-producing bacteria can shape the microbial environment by favoring microorganisms, which directly use acetate as substrate and produce other beneficial metabolites such as butyrate [2,14,15]. Acetate has several health benefits for the host, connected to intestinal mucus production [15], metabolism [16], and the immune system [17]. It has been shown that acetate modulates the function of CD8^+^ T cells through enhanced production of interferon gamma in cancer [18] or an optimized immune response after a systemic bacterial infection [19]. Acetate, produced by gut bacteria, could target other immune cells, like IgA-producing B cells [20], neutrophil activity [21], and CD4^+^ T cells [22]. As an actor on innate and acquired immunity, acetate modulates defenses against pathogens [23]. During an influenza infection, modulation in gut microbiota’s composition occurs with a resulting drop in SCFAs [24,25]. We have shown that treatment with acetate reduces enteric *Salmonella enterica* serovar Typhimurium (*S.t*) and pulmonary *Streptococcus pneumonia* (*S.p*) bacterial superinfection post-influenza [25,26]. In this setting, acetate targets the free fatty acid receptor 2 [25]. Recent findings indicated that gut microbiota alteration, with a low amount of acetate production, can favor colonization and systemic dissemination of *S.t* [24,25].

Regarding the beneficial and systemic effects of acetate, we intended to isolate a new commensal strain to develop new probiotics potentially exploitable in the context of respiratory infections. We designed a framework to obtain an intestinal high-acetate-producing commensal strain, and we challenged the effect of the selected future probiotic strain in vitro and in two preclinical influenza models with *S.p* and *S.t* superinfection.

## 2. Material and Methods

### 2.1. Isolation of Extremely Oxygen-Sensitive Strains

Extremely Oxygen-Sensitive (EOS) bacteria have been isolated from 1 g of feces of a healthy human donor with the method described in Martin et al. [2] with the difference that in this study, the culture medium was BHIS (brain-heart infusion medium supplemented with 0.5% yeast extract (Difco, Franklin Lakes, NJ, USA), cellobiose (1 mg/mL, Sigma, Angers, France), maltose (1 mg/mL, Sigma), and cysteine (0.5 mg/ml, Sigma, Angers, France) (further in the article mentioned as “BHIS(+)”). Strains that could not be cultured again after being frozen (−80 °C in 16% glycerol) were further eliminated.

### 2.2. Blautia Strains Used

For this study, *Blautia hansenii* DSM20583 was purchased from Leibniz Institute DSMZ-German Collection of Microorganisms and Cell Cultures. The *Blautia faecis* strain, identified in this work, has been deposited at the Leibniz Institute DSMZ-German Collection of Microorganisms and Cell Cultures under the reference *Blautia* DSM33383 and patented under the application WO2021/151956.

### 2.3. Culture Conditions for Blautia hansenii DSM20583 and Blautia faecis DSM33383

*Blautia* strains were grown in liquid BHIS(+) medium inside an anaerobic chamber (N_2_: 90%, CO_2_: 5% and H_2_: 5%). The bacterial cultures were introduced into the anaerobic chamber in frozen state and brought into culture at a 10% or 1% inoculation in liquid BHIS (+) for 24 h at 37 °C. Then, 1% inoculations were made for overnight (O/N) growth (16–18 h) at 37 °C. Bacterial growth of a 2% inoculation from the O/N culture was followed by taking the OD at 600 nm every hour until the maximum OD_600nm_ was reached. Growth rates (µ_max_) were calculated from the linear part of the logarithmic transformed OD_600nm_.

### 2.4. Supernatant Collection from Bacterial Cultures

Supernatants from the stationary phase were collected by centrifugations (2 × 10 min at 5000× *g* at 4 °C) outside of the anaerobic chamber and stored at −20 °C before use for SCFA analysis and HT-29 cell co-incubation tests.

### 2.5. 16S rDNA Identification of EOS Strains by PCR from Pure Colony

For the screening of the isolated bacterial strains, 16S rDNA was sequenced by using the primers fD1 (^5′^AGAGTTTGATCCTGGCTCAG^3′^) and rp2 (^5′^ACGGCTACCTTGTTACGACTT^3′^) and Dreamtaq polymerase (Thermofisher, Vilnius Lithuania) for amplification of DNA from a pure colony. Conditions were 95 °C for 15 min, followed by 38 cycles of 95 °C for 30 s, annealing at 52 °C for 30 s, and extension at 72 °C for 1.5 min, with a final extension at 72 °C for 10 min for bacterial amplification and sequencing by Eurofins (Ebersberg, Germany). The 16S rDNA sequences were compared using the NCBI blast tool (https://blast.ncbi.nlm.nih.gov/Blast.cgi, accessed on 1 January 2020) in order to identify the taxonomic assignment to which they belong.

### 2.6. Genomic DNA Extraction of Blautia faecis DSM33383 16S rDNA for Sequencing and Phylogenetic Analysis

For genomic DNA extraction of *Blautia faecis* DSM33383, the strain was grown in several replicates until the stationary phase (10 to 18 h). Subsequently, 2–3 mL per culture was taken and centrifuged at 5000× *g* for 10 min at 4 °C. Genomic DNA was extracted following three steps: cell lysis, protein and RNA denaturation using chemical methods, DNA purification using QIAGEN Genomic-tip 100/Gn, and DNA precipitation using the isopropanol method.

These samples were pooled (total of 5.3 × 10^8^ CFU), and cell pellets were washed with TRIS-EDTA buffer, bacterial cell lysis was performed with lysozyme 25 µL (200 mg/mL) for one hour at 37 °C, RNase and proteinase K were added to the lysis solution, and incubation was performed for 30 min at 37 °C. Genomic DNA was further purified using a QIAGEN Genomic-tip 100/G and proteinase K, according to the manufacturer’s protocol (QIAGEN-Genomic-DNA-Handbook).

For a higher quality sequence for the isolated *Blautia faecis* strain primers F: ^5′^ACTTTATCAGAGAGTTTGAT^3′^ and R: ^5′^TAGAAAGGAGGTGATCCAGC^3′^ were selected to extract the full 16S rDNA gene sequence from the gDNA, which was sequenced by Eurofins (Ebersberg, Germany) (Appendix A).

A phylogenetic tree was constructed using the phyML tool (https://ngphylogeny.fr/, accessed on 1 January 2020 ) by selecting the Gblocks program to eliminate poorly aligned position and divergent regions [27]. A total of 158 16S rDNA sequences of species belonging to the *Blautia* genus (including 132 sequences from the IMG/M database, 17 from the NCBI database, and 3 from the PATRIC database) were used, as well as the 3 species, *Faecalibaterium prausnitizii* A2-165, *Bacillus cereus* E33L, and *Bacillus subtilis* sp. *subtilis* 168, leading to the rooting for the phylogenetic analysis. The phylogenetic tree result was visualized after 500 bootstrap repetitions by the neighbor-joining algorithm.

### 2.7. Measurement of SCFA Concentrations in Bacterial Supernatant

Acetate, butyrate, and propionate were quantified in the bacterial supernatant using gas chromatography (Agilent Technologies, Les Ulis, France) [28]. 2-Ethylbutyrate (Sigma, Angers, France, 2-ethylbutyric acid, 99%) at a concentration of 20 mM was used as internal standard in a 1:4 ratio to the bacterial supernatant for normalization of the data per run. An external standard (Sigma, Angers, France, volatile free acid mix) was used for identification and quantification of acetate, butyrate, and propionate in the samples in each run. Analyses were made using the OpenLab Chemstation software (Agilent, Les Ulis, France). Concentrations are given in mM.

### 2.8. Epithelial Cell Culture and In Vitro Anti-Inflammatory Assays

In vitro anti-inflammatory assays were performed by co-incubation of filter-sterilized bacterial supernatant collected in the late stationary phase (after 24 h of growth) and human colorectal adenocarcinoma cell line (HT-29, ATCC HTB-38). The protocol used for the co-incubation experiments with HT-29 cells was adapted from Kechaou et al. [29].The HT-29 cells were cultured in 24-well plates with Dulbecco’s Modified Eagle Medium (DMEM) with 1% glutamine, supplemented with 5% fetal calf serum (Ref. CVFSVF00-01, Eurobio). For the assay, cells were challenged with recombinant human tumor necrosis factor alpha (TNFα) (5 ng/mL). Each well was then supplemented with 25% of either the bacterial supernatant (i.e., *Blautia faecis* DSM33383 or *Blautia hansenii* DSM 20583), or with the control BHIS(+) medium (without bacterial growth). Every control and sample was tested in triplicate, with 6 biological replicates for the groups with bacteria and 3 biological replicates for the group with BHIS medium supplemented with acetate. The cells were incubated at 37 °C with 10% CO_2_ for 6 h. The supernatant of these cells was collected and used directly for a Lactate Dehydrogenase Activity (LDH) assay (CytoTox 96^®^ Non-Radioactive Cytotoxicity Assay, Promega) to test if the cell membranes were still intact after the treatment. The rest of the cell supernatant was stored at −80 °C until IL-8 quantification.

### 2.9. IL-8 Cytokine Quantification

IL-8 cytokine release of the co-incubated HT-29 cells was quantified using human IL-8 ELISA max standard kits (Biolegend, Paris, France). Results from cells treated with bacterial supernatants were normalized with results from cells treated with BHIS(+) medium.

### 2.10. Mice and Ethics Statement

Specific pathogen-free C57BL/6J mice (7-week-old, male) were purchased from Janvier (Le Genest-St-Isle, France). Mice were maintained in a biosafety level 2 facility in the Animal Resource Center at the Lille Pasteur Institute for at least two weeks prior to usage to allow appropriate acclimatization. Mice were fed a standard rodent chow (SAFE A04, SAFE, Augy, France) and had access to water ad libitum. This diet contained ~11.8% fiber, including ~10% water-insoluble fiber (3.6% cellulose) and 1.8% water-soluble fiber. 

All experiments complied with current national and institutional regulations and ethical guidelines (Institut Pasteur de Lille/B59-350009). The protocols were approved by the institutional ethical committee “Comité d’Ethique en Experimentation Animale Nord-Pas de Calais (CEEA-075)”. The study was authorized by the “Education, Research and Innovation Ministry” under registration number APAFIS#13743-2018022211144403 v2.

### 2.11. Infections Gastric Administration and Assessment of Bacterial Loads in Mice

The mouse-adapted H1N1 A/California/04/2009 (pdm09), H3N2 IAV (influenza A Virus) strain Scotland/20/1974, *S. pneumoniae* serotype 1 (S.p), and *Salmonella enterica* serovar Typhimurium (*S.t*) were described in [25,26].

For infection with IAV alone, mice were anesthetized by intraperitoneal injection of 1.25 mg of ketamine plus 0.25 mg of xylazine in 100 µL of phosphate buffered saline (PBS), and then intranasally (i.n.) infected with 50 µL of PBS containing (or not, in a mock sample) 100 plaque forming units (PFU) of H1N1 A/California/04/2009 (pdm09) or 50 PFU of H3N2 IAV strain Scotland/20/1974 [30,31]. For superinfection with *S. pneumoniae*, mice were anesthetized after 8 days of H1N1 A/California/04/2009 infection and infected with 30 µL containing 1 × 10^3^ CFU. This dose was sufficient to allow bacterial outgrowth, dissemination, and death in mice previously infected with IAV. In double infected mice, bacteria in the lungs were counted 30 h after the bacterial challenge by plating serial 10-fold dilutions of lung homogenates onto blood agar plates. The plates were incubated at 37 °C with 5% CO_2_ overnight, and viable bacteria were counted 24 h later. Survival and body weight were monitored daily after IAV infection, and mice were euthanized when they lost in excess of 20% of their initial body weight. For secondary infection with *Salmonella enterica* serovar Typhimurium (*S.t*), H3N2 IAV strain Scotland/20/1974-infected mice were intragastrically challenged at 7 d.p.i. with (S.t), (1 × 10^4^ CFU, 200 µL). This dose was sufficient to allow bacterial outgrowth, dissemination, and death in mice previously infected with IAV. Survival and body weight were monitored daily after IAV infection, and mice were euthanized when they lost in excess of 20% of their initial body weight. To study the beneficial effect of *Blautia faecis* DSM33383 or “F2 strain”, O/N cultures were prepared and instantly frozen in liquid nitrogen and stored in BHIS(+) and 16% glycerol in −80 °C until the day of administration to the mice. The CFU was determined after the freezing process. As control, BHIS(+) and 16% glycerol were frozen in the same way.

Mice were treated daily intragastrically with BHIS(+) medium containing or not, in a control sample (vehicle, vh), DSM33383 strain (5 × 10^6^ CFUs/200 µL) or F2 strain (2 × 10^7^ CFUs/200 µL) at 1 d.p.i. until 7 d.p.i.

### 2.12. Measurement of SCFA Concentrations in Cecal Content of Mice

SCFAs were quantified in the cecal content after extraction with diethyl ether using GC-2014 gas chromatography with AOC-20i auto injector (Shimadzu’s, Hertogenbosch, the Netherlands). Results are expressed as µmol/g of cecal content.

### 2.13. RNA Manipulation Procedures and Real-Time Quantitative RT-PCR

Total RNA from lungs tissues were extracted with a NucleoSpin^®^ RNA kit (Macherey-Nagel, Hoerdt, Germany). RNA was reverse-transcribed with a High-Capacity cDNA Archive Kit (Life Technologies, Courtaboeuf, France). The resulting cDNA was amplified using SYBR Green-based real-time PCR and QuantStudio™ 12K Flex Real-Time PCR Systems (Applied Biosystems™, Courtaboeuf, France) following the manufacturer’s protocol. Relative quantification was performed using the gene coding glyceraldehyde 3-phosphate dehydrogenase (*gapdh*). Specific primers were designed using Primer Express software (Applied Biosystems, Villebon sur Yvette, France) (Appendix A). Relative mRNA levels (2^−ΔCt^) were determined by comparing (a) the PCR cycle thresholds (Ct) for the gene of interest and the housekeeping gene *Gadph* (ΔCt) and (b) ΔCt values for treated and control groups (ΔCt). Data are expressed as a fold-change over the mean gene expression level in mock-treated mice.

### 2.14. Statistical Analysis

Results are expressed as the median ± interquartile range (IQR) unless otherwise stated. All statistical analyses were performed using GraphPad Prism v8 software. A Mann–Whitney U test was used to compare two groups unless otherwise stated. Comparisons of more than two groups with each other were analyzed with Kruskal–Wallis tests with Dunn’s multiple comparisons tests. Results were found significant if *p* < 0.05 (*), *p* < 0.01 (**), *p* < 0.001 (***), or *p* < 0.0001 (****).

## 3. Results

### 3.1. The Isolation of Extremely Oxygen-Sensitive Strains and Identification of a High-Acetate-Producing EOS Strain-Blautia faecis DSM33383

As described in Figure 1, screening of Extremely Oxygen-Sensitive (EOS) strains from a fecal sample of a healthy human donor were performed according to four criteria: (i) on their ability to grow only under strict anaerobic conditions; (ii) on their cell viability after a freezing process; (iii) on their inability to induce proinflammatory cytokine interleukin (IL)-8 production in HT-29 cells; and finally, (iv) on their ability to produce acetate. From the fecal bulk, 144 isolates were cultured anaerobically in BHIS(+). Among these isolates, we excluded those that were able to survive on plates for more than 20 min in the presence of oxygen. The remaining 21 isolates were then challenged to be re-cultured after freezing, losing three strains. Of 18 remaining isolates, 14 were selected, because they did not induce overproduction of IL-8 in HT-29 cells (data not shown). Among these 14 strains, some produced acetate, butyrate, or both acetate and butyrate as found in samples from the stationary phase (Table 1). None of the strains produced a detectable level of propionate (<0.2 mM). Of the 11 acetate-producing strains, there were 10 strains that were able to produce a small amount of acetate (<10 mM) and one strain which was a high acetate producer (31 mM). In addition, for these 14 strains, taxonomic assignment based on their partial or complete 16S rDNA sequence was performed using the NCBI blast tool (https://blast.ncbi.nlm.nih.gov/Blast.cgi, accessed on 1 January 2020) (Table 1).

In order to refine the species name of the highly acetate-producing DSM833383 strain, a phylogenetic analysis based on partial and complete 16S rDNA sequences of the genus *Blautia* in the IMG M, NCBI, and PATRIC genomic databases in May 2021 were performed. A total of 158 rDNA sequences from a high-quality control and representative of the different clusters of the genus *Blautia* were selected for this study. As was shown in Figure 2 in the phylogenetic tree of the genus *Blautia*, the 16S rDNA sequence of the DSM33383 strain (Appendix A) was affiliated with the *Glucerasea/faecis* cluster. The result of the two-to-two sequence alignment of the DSM833383 strain with the four strains of the species *Blautia glucerasea* available in the IMG/M databases revealed 95.6–98% identity, while with the three strains of the species *Blautia faecis* available in PATRIC databases, indicated 99.4–99.7% identity. *Blautia faecis* DSM33383 was affiliated to the *Lachnospiraceae* family in the *Firmicutes* phylum. Thus, we have identified a new strain of the species *Blautia faecis* DSM33383.

### 3.2. The Characteristics of Blautia faecis DSM33383

The *B. faecis* DSM33383 strain reached a maximum OD 600 nm of 3.0 with a growth rate of 0.55 (Figure 3A,B) and produced 31.4 ± 5.6 mM of acetate (Figure 3C) in BHIS(+). The IL-8 response of TNFα-stimulated HT-29 cells was significantly reduced when the supernatant of *B. faecis* DSM33383 was added to the cell medium, compared to that with the addition of BHIS(+) without bacterial supernatant (Figure 3D). When considering another commercially available strain belonging to the *Blautia* genus, *B. faecis* DSM33383 had acetate production superior to that of the commercial strain *Blautia hansenii* DSM20583, which displayed a nonsignificant reduction of IL-8 production (Appendix A).

All of the experimental conditions showed a similar release of LDH as the negative control with DMEM medium, which means that none of the conditions were cytotoxic to the cells (Appendix A).

The *Blautia faecis* DSM 33383 strain was selected for further study in a mouse model.

### 3.3. Blautia faecis DSM33383 Administration during Influenza Protects against Lung Bacterial Superinfection with Streptococcus pneumoniae

We investigated the potential beneficial effect of *Blautia faecis* DSM33383 intragastrical administration on bacterial superinfection. Indeed, our recent data indicate that the drop of SCFA production, mainly acetate, during influenza contributes to enhanced susceptibility to secondary bacterial infection [25]. *Streptococcus pneumoniae* (*S.p*) was used as a relevant bacteria as it is the first cause of bacterial pneumonia worldwide [32]. To this end, IAV-infected mice were treated daily with *Blautia faecis* DSM33383 at day 1 post-IAV infection until 7 days post-infection (d.p.i.). IAV-infected mice were superinfected at 8 d.p.i., the peak of susceptibility to secondary bacterial infection (Figure 4A). Remarkably, oral administration of *Blautia faecis* DSM33383 lowered the bacterial load in the lungs of double-infected mice (Figure 4B. It also affected the spreading of bacteria from the lungs as revealed by lower bacterial counts in the spleen, relative to that of the control (Figure 4B). Of interest, *Blautia faecis* DSM33383 gavage significantly improved mouse survival (~50% relative to ~80% in the control group) (Figure 4C). This effect associated with a lower weight loss post-influenza and with a more rapid weight regain of double-infected mice (Figure 4C). *Blautia faecis* DSM33383 administration also associated with reduced expression of genes encoding inflammatory cytokines such as *Tnfα*, *Il1β*, *Oas3*, and *Isg15* (Appendix A). When the *Coproccus comes* F2 strain (Table 1) was administrated following the same protocol of gavage, no protective effects were observed (Appendix A). Taken as a whole, *Blautia faecis* DSM33383 administration significantly reduced the outcomes of post-influenza bacterial superinfection in the lung.

### 3.4. Blautia faecis DSM33383 Administration Tends to Reduce Secondary Salmonella enterica *serovar Typhimurium* Infection Post-Influenza

We then investigated the potential beneficial effect of *Blautia faecis* DSM33383 administration on secondary bacterial enteric superinfection. *Blautia faecis* DSM33383 administration tended to lower the mortality rate of double-infected animals (~30% survival rate relative to ~12% survival rate in the control group (Figure 5A)). Weight curve analyses revealed that *Blautia faecis* DSM33383 administration tended to ameliorate weight regain after enteric superinfection (Figure 5B). Hence, DSM33383 administration during the course of influenza tended to reduce secondary enteric infection.

## 4. Discussion

The concept of gut–lung connection was born out of the observation that different lung acute or chronic diseases can be modulated by changes in the intestinal microenvironment [33,34,35,36]. We have also shown in mice that the perturbation of the gut microbiota during an IAV infection might favor respiratory bacterial superinfection [25,26]. Altered gut–lung communication in old people renders the elderly more sensitive to infections [37]. The existence of a gut–lung axis justifies the use of intestinal probiotics to manage respiratory health, as it has been proposed in the respiratory syncytial virus model [38]. In the current study, a variety of bacterial strains have been isolated from a human fecal sample, and *Blautia faecis* DSM33383 was selected because it was a strict anaerobic, high-acetate-producing species which showed a reducing effect on in vitro IL-8 production induced by treatment with TNFα. Moreover, a reduced inflammatory response was also found in influenza-infected mice suffering from an *S.p* superinfection when treated intragastrically with *Blautia faecis* DSM33383 but not with *Coprococcus comes* F2 strain. The administration of *Blautia faecis* DSM33383 led to a reduction of the bacterial load in the lungs and to an enhanced survival rate. Moreover, although less marked, *Blautia faecis* DSM33383 gavage also tended to partial protection upon secondary enteric *S.t* infection. The limitations of preclinical models obviously render the results difficult to translate in humans. However, our results are encouraging to further study bacterial preventive strategy against lung infections.

The supernatant of *Blautia faecis* DSM33383 caused a reduction of TNFα-stimulated IL-8 production in the intestinal epithelial cell line used in this study. We can suppose that *Blautia faecis* DSM33383 produced another extracellular metabolite with an anti-inflammatory effect regarding IL-8 production. Like the MAM peptide of *Faecalibacterium prausnitzii* [7], this could be an anti-inflammatory molecule specific to the *Blautia faecis* DSM33383 strain. For example, some studies have shown that one strain from *Blautia sp.* was able to produce anticancer and anti-inflammatory metabolites through demethylase of polymethoxyflavones and curcuminoids [39,40,41]. Whatever the underlying mechanisms of action, an anti-inflammatory effect due to *Blautia faecis* DSM33383 was also observed in the mouse model. *Blautia faecis* DSM33383 did not have an effect on the influenza infection itself but was able to reduce disease severity and bacterial load in mice superinfected with *S. pneumoniae*. The mechanisms behind these protective effects are still elusive. It is likely that acetate remotely arms alveolar macrophages to control an *S. pneumoniae* infection, as our prior findings indicated [25]. Other components emanating from *Blautia faecis* DSM33383 or other commensals (supposedly impacted by the treatment) might also play a role. Analysis of the gut microbiota composition in mice infected with influenza and receiving or not *Blautia faecis* DSM33383, might be indicative and should be done in future experiments.

It is quite likely that our framework of selection excluded other potentially beneficial strains. This is in part because certain strains have specific nutritional and environmental needs, which are difficult to reproduce in vitro. When using in vitro cell line models to study anti-inflammatory effects of EOS strains, live bacteria cannot be tested, because of the anaerobic nature of these strains, and a compromise must be made to use only the supernatant of the strains after being cultured under anaerobic conditions. The limitation, on the other hand, was also due to limited capacity to test in detail the functional and physiological capacities of all the strains, especially with an in vivo mice model, limiting the number of experiments. That is why not all strains that were isolated during the course of this study could be tested. An experimental design that overcomes the limitation in quantity and ethics of the in vivo models and that would resemble the natural anaerobic environment of the intestines more than the in vitro cell line models would be that of organoids. There have been studies that report the successful injection of a supernatant form anaerobically grown bacterial cultures into organoid tissues to study the host–microbe interactions ex vivo [42,43]. Moreover, complex microbiota communities from fecal samples have been successfully injected and maintained alive within organoids [44]. This makes organoids a promising alternative for in vitro and in vivo models, regarding the study of strictly anaerobic *Blautia faecis* DSM33383.

Several *Bifidobacterium* strains are also found to produce acetate, be anti-inflammatory, and protect the host from pathogenic infections [45]. However, *Bifidobacteria* presence among the microbiota in the gut differs throughout the life of the host [8]. When administered for probiotic use, several *Bifidobacteria* strains do not consistently or at all colonize the gut of healthy adults [46,47,48]. The *Blautia* genus is abundant in the human gut microbiota and has high tropism for the intestinal tract. Other *Blautia faecis* strains and also other *Blautia* species like *Blautia wexlerae* can be found in mouse fecal matter after fecal microbiota transfer from a healthy human adult [49]. Since different members of the genus seem to display different levels of colonization, we have observed an increased *Blautia* count after specific qPCR amplification of DNA from fecal samples of mice after daily intragastric administration of *Blautia faecis* DSM33383 for ten days (data not shown, preliminary data).

For the future development of commercial products, a *B. faecis* strain like DSM33383 should not be considered as a conventional probiotic, which usually belongs to the genera *Bifidobacterium*, *Lactobacillaceae,* other lactic acid bacteria and a couple of exceptions [50]. If we choose to develop nutritional products, *Blautia* should be considered a Novel Food because *Blautia* is absent from the QPS-list EU regulation (EU 2015) or GRAS-list, and we cannot document its history of safe use before 1997 [51]. To state the safety, a convincing demonstration and arguments are required to convince the regulatory authorities with a critical genome analysis and phenotypic functional assays. Then, of production of the strain with industrial constraints could be scaled up [52]. The use of a novel bioreactor dedicated to the culture of EOS strains, which is currently being tested, could be a first step in scaling the production up (data not shown). We can also foresee that *Blautia faecis* DSM33383, being able to grow in different media (data not shown), should accommodate to a medium that complies with the Good Manufacturing Practices for industrial use. To extend the use of *Blautia* as a novel food associated with a health claim, the demonstration of a causal link between the use of *Blautia* and its beneficial effect should be convincing with other preclinical models and clinical trials in humans. Besides the nutritional strategies, *Blautia faecis* DSM33383 could be developed as a therapeutic drug, either through its supernatant, isolated bioactive compounds, or as a Live Biotherapeutic Product [53]. To reach this pharmaceutical status, the requirements are also based on further characterization, safety concerns, and more preclinical and clinical evidence [54].

## Figures and Tables

**Figure 1 nutrients-14-01478-f001:**
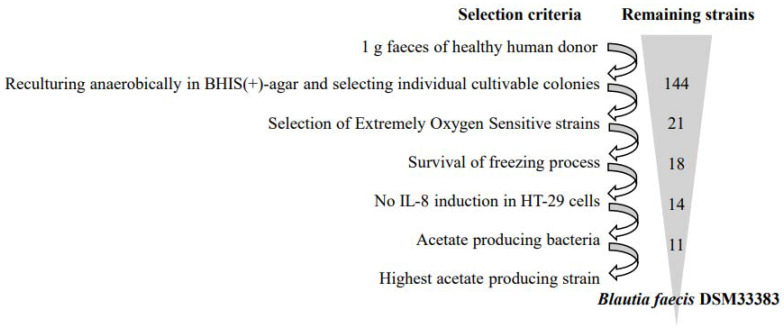
Overview of selection steps leading to the isolation of acetate-producing EOS strains.

**Figure 2 nutrients-14-01478-f002:**
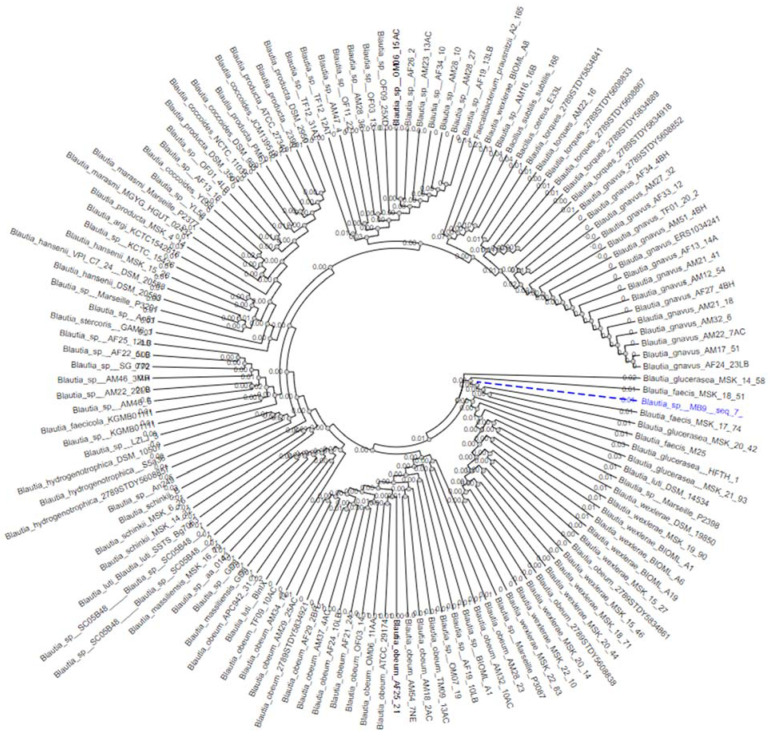
Phylogenetic tree of *Blautia* species from IMG, NCBI, and PATRIC databases.

**Figure 3 nutrients-14-01478-f003:**
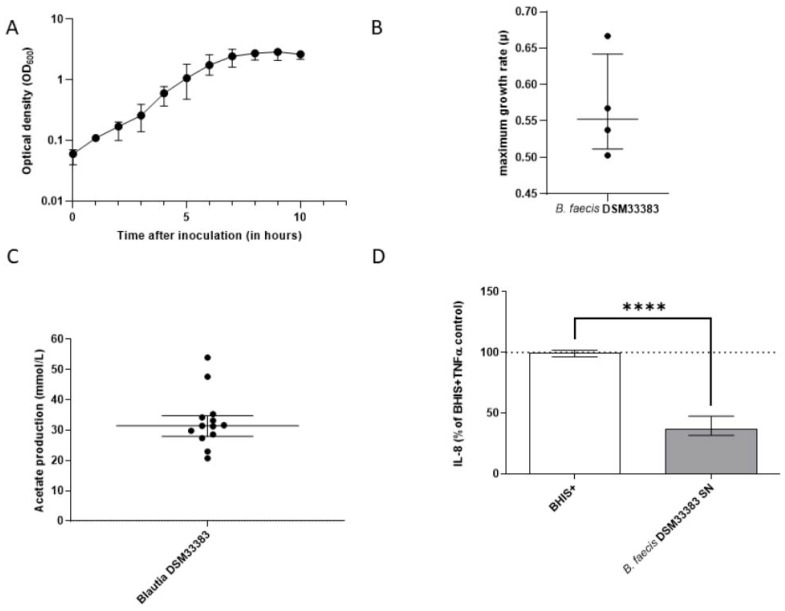
Growth characteristics of *B. faecis* DSM33383 in BHIS+ medium and in vitro anti-inflammatory effect. (**A**) Kinetic curve of *B. faecis* DSM33383 grown in BHIS+. Logarithmic representation of the OD at 600 nanometers (nm), taken with one-hour intervals. *n* = 4. Median with interquartile range (IQR). (**B**) Maximum growth rates (µ) of the selected growth curves. *n* =4. Median with interquartile range (IQR). (**C**) Acetate production in mM by *B. faecis* DSM33383, sampled in the stationary phase. Acetate levels of each sample were normalized with the acetate level of the BHIS+ medium without bacteria. *n* = 13 for *B. faecis* DSM33383. Median with interquartile range (IQR). (**D**) The IL-8 response of TNFα stimulated HT-29 cells supplemented with the culture supernatant (SN) (25%) of either *B. faecis* DSM33383 compared to the control with 25% BHIS+. *n* = 6 Median with interquartile range (IQR). **** *p* < 0.0001.

**Figure 4 nutrients-14-01478-f004:**
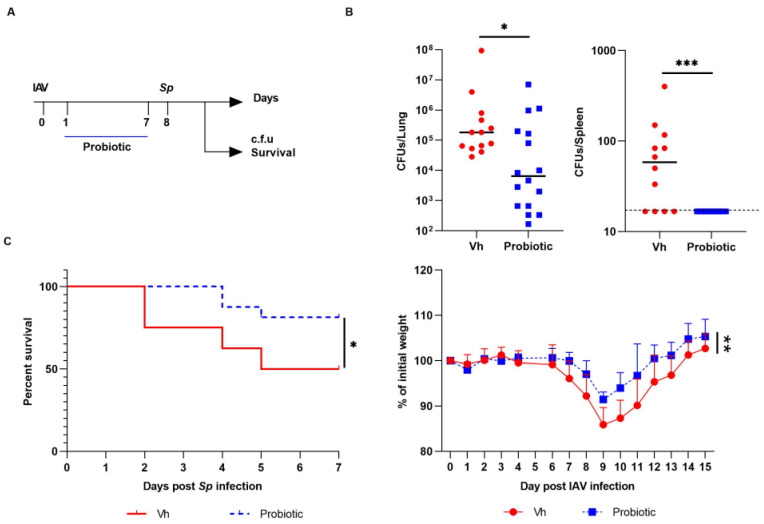
Effect of *B. faecis* DSM33383 supplementation during IAV infection on secondary *Streptococcus pneumoniae* infection. (**A**) Schematic representation of the double infection system. IAV-infected mice (8 d.p.i.) were infected with *S.p* (1 × 10^6^ CFU). IAV-infected mice were gavaged or not with *B. faecis* DSM33383 (5 × 10^6^ CFU/200 µL) at 2 d.p.i. until 7 d.p.i. (**B**) The number of bacteria was determined in lung (left) and spleen (right) 30 h after the bacterial challenge (*n* = 13–16, two pooled experiments shown). (**C**) The survival (left) and body weight evolution (right) (in % initial body weight) of doubly infected animals, treated or not, were monitored (*n* = 16, two pooled experiments shown). Significant differences were determined using the Mann–Whitney *U* test (**B**, **C**). Mice survival was compared using Kaplan–Meier analysis and the log-rank test (**C**) (* *p* < 0.05; *** *p* < 0.001, Vh: vehicle, IAV: influenza A Virus.).

**Figure 5 nutrients-14-01478-f005:**
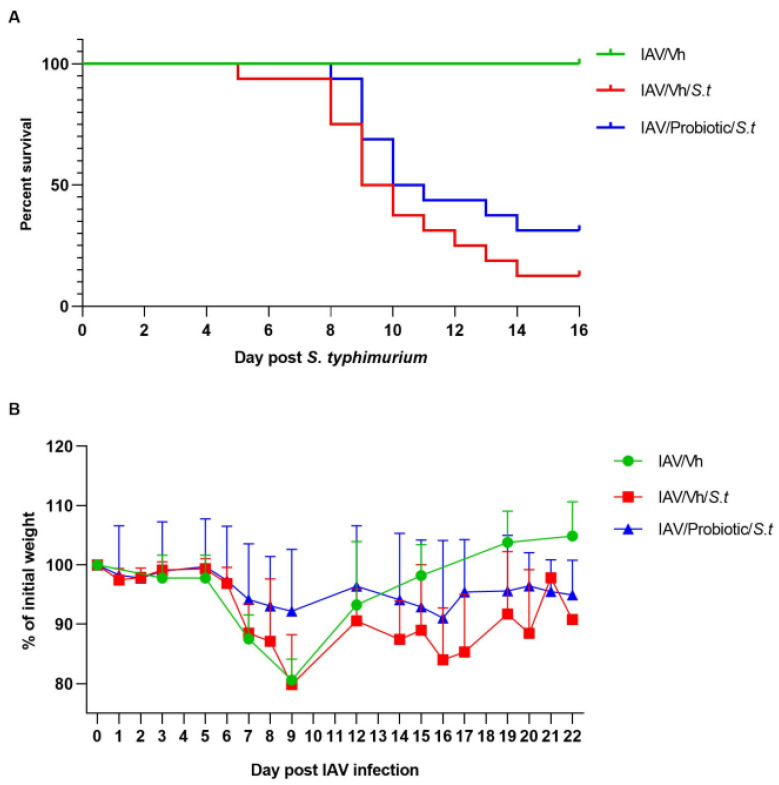
Effect of *B. faecis* DSM33383 supplementation during IAV infection on secondary *Salmonella* infection. IAV-infected mice (7 d.p.i.) were infected with *Salmonella enterica* serovar Typhimurium (*S.t*) (1 × 10^4^ CFU). IAV-infected mice were gavaged or not (Vh) with *B. faecis* DSM33383 (5 × 10^6^ CFU/200 µL) at 2 d.p.i. until 7 d.p.i. (**A**) The survival and (**B**) body weight evolution (right) (in % initial body weight) of doubly infected animals, treated or not, were monitored (*n* = 16, two pooled experiments shown). IAV, influenza A Virus. Vh: vehicle.

**Table 1 nutrients-14-01478-t001:** Closest homology (%) to the 16S of the 14 selected strains, with the 16S rDNA sequences of the strains in the NCBI database. The homology was based on the partial or full 16S rDNA of strains isolated from a healthy human fecal sample. Short chain fatty acid production in stationary phase after an overnight culture in BHIS(+). Median ± interquartile range, *n* = 4 except for strains E5, MA9, and MC4, where *n* =1, nd = not detected (under 0.2 mM).

Strain	16S rDNA NCBI Blast	Acetate Production (mM)	Butyrate Production (mM)	Propionate Production (mM)
C7	*Ruminococcus sp.* (99%)	4.9 ± 2.9	nd	nd
D4	*Coprococcus eutactus* (99%)	3.4 ± 1.5	6.6 ± 3.3	nd
E5	*Ruminococcus albus* (94%)	4.7	nd	nd
E9	*Fusicatenibacter saccharivorans* (99%)	8.2 ± 5.2	nd	nd
E12	*Eubacterium ventriosum* (99%)	2.3 ± 1.5	7.5 ± 2.6	nd
F2	*Coprococcus comes* (99%)	7.2 ± 4.9	6.5 ± 3.9	nd
MA4	*Anaerostipes hadrus* (99%)	nd	16.7 ± 6.6	nd
MA8	*Fusicatenibacter saccharivorans* (99%)	4.8 ± 8.3	nd	nd
MA9	*Ruminococcus sp.* (97%)	6.4	nd	nd
MA11	*Roseburia faecis* (99%)	nd	11.2 ± 4.4	nd
MB6	*Eubacterium ventriosum* (99%)	1.8 ± 0.7	5.2 ± 2.4	nd
DSM33383	*Blautia sp* (96%)	31.3 ± 6.2	nd	nd
MC1	*Eubacterium ventriosum* (98%)	1.6 ± 2.0	6.5 ± 4.5	nd
MC4	*Roseburia sp.* (91%)	nd	10.6	nd

## Data Availability

The authors confirm that the data sets are available in the “data INRAE” repository platform accessible through doi.org/10.15454/T9X2V0. The data and calculation deposited are growth curves and rates of two Blautia strains, short chain fatty acids of all strains, IL8 elisa and LDH outputs, phylogeny of 16S blautia strains, individual weight of mice, and CT values RT-PCR. The complete 16S rDNA gene of *Blautia faecis* DMS33383 has been deposited in GenBank under the accession number OL782608.

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
