# Peer review of "Description of a Newly Isolated Blautia faecis Strain and Its Benefit in Mouse Models of Post-Influenza Secondary Enteric and Pulmonary Infections"

_nutrients, 2022, doi:10.3390/nu14071478_

Round 1
Reviewer 1 Report
Dear authors,
I enjoyed reading the manuscript very much.
The manuscript is written very good. It is very interesting covering the isolation of Blautia faecis DSM33383 strain from a stool sample of healthy donor. According to the results of the study this strain might be a promising candidate for the treatment of respiratory infectious diseases as it proved to be beneficial in mice model of influenza and also reducing the secondary Salmonella enterica serovar Thyphiurium infection.
The data presented in tables and figures are easy to understand and are clearly presented.
The literature used is up to date.
There is only one typo that needs correction - please check line 49 - the cited literature in the brackets in by accident put in the middle of the word panel.
I am looking forward to reading more about your research.
With kind regards
Reviewer 2 Report
This is rather a curious paper: a very well planned and performed in its first part describing steps leading to isolation and characterization of the new strain of Blautia faecis and its role in production of acetate and other free fatty acids etc. Unfortunately, the second part presenting experiments arranged to demonstrate its functions as alimentary “protector?” of the bacterial superinfections in the course of influenza, or even pneumococcal pneumonia are not correctly designed. Moreover, the conclusions go certainly too far and represent rather good wishes than well substantiated data derived from the study.
The animal experiment presented in this paper is based on a simple assumption that influenza virus infected mice can mimic natural infection in humans and that S.pneumoniae is the main pathogen involved in post-influenza bacterial pneumonia. Also type 1 influenza virus is reported as not very well adapted to mice models and rarely used. According to epidemiological data, S.pneumoniae is rather a rare bacterium causing bacterial superinfection in last few outbreaks and endemic cases; More common and importat are S.pyogenes, H.influenzae and Gram-negative rods. It is commonly accepted that mechanisms predisposing to bacterial superinfections in influenza are related to altered "innate-like" unconventional T cells antibacterial activity but not to free fatty acids deficiency. The last mechanism is attributed to influenza but not to post-influenza bacterial superinfections.
The most important failure is a total lack of the appropriate controls in the animal experiments demonstrating that the effect attributed to B.faecis supplementation is not just accidental and that no other bacteria used as a control not shown a similar effect. This is even more important in the situation in which the effects of the B.faecis supplementation were measured by death rates and weight loss only. There are very recent reports in medical literature on positive effect of the traditional probiotics on bacterial pneumonia in ventilated patients in relation to Covid-19 infection and other virus infections. Moreover, the authors ideas that oral application of the B.faecis will be effective as prophylaxis of the bacterial superinfections in influenza are not sufficiently supported. The medical standards of the prophylaxis and treatment of influenza are stable and universal since many years, and they are based on highly effective vaccines and antiviral drugs. The same is true for pneumococcal pneumonia which is prevented by polysaccharide vaccines and treated with properly selected antibiotics. Why and what for an approach based on dietary supplementation with a bacterium producing acetate as the most important mechanism should be developed, commertialized and used in these infections?
Minor remark: The authors should not use term “treatment” which should be secured for medicinal products only.
Author Response
Reviewer 2 :
This is rather a curious paper: a very well planned and performed in its first part describing steps leading to isolation and characterization of the new strain of Blautia faecis and its role in production of acetate and other free fatty acids etc. Unfortunately, the second part presenting experiments arranged to demonstrate its functions as alimentary “protector?” of the bacterial superinfections in the course of influenza, or even pneumococcal pneumonia are not correctly designed. Moreover, the conclusions go certainly too far and represent rather good wishes than well substantiated data derived from the study.
The animal experiment presented in this paper is based on a simple assumption that influenza virus infected mice can mimic natural infection in humans and that S.pneumoniae is the main pathogen involved in post-influenza bacterial pneumonia. Also type 1 influenza virus is reported as not very well adapted to mice models and rarely used.
Answer : We have used a mouse-adapted strain as it is indicated line 197 in 2.11 section of Material and Methods. Regarding, the model, we have now precised that it has limitations regarding the human situation (yellow track sentences at the beginning of the discussion). “The limitations of preclinical models obviously render the results difficult to translate in Humans. But, our results are encouraging to further study bacterial preventive strategy against lung infections”.
According to epidemiological data, S.pneumoniae is rather a rare bacterium causing bacterial superinfection in last few outbreaks and endemic cases; More common and importat are S.pyogenes, H.influenzae and Gram-negative rods.
Answer : With all due respect, we do not agree with this comment. Infection with S. pneumonia is indeed a major cause of secondary bacterial infections post-influenza (i.e. the review from Mc Cullers, who is a world-class specialist :Mc Cullers (2014). Nat. Rev. Microbiol.12:252 (doi: 10.1038/nrmicro3231) reference 32 in the text.
It is commonly accepted that mechanisms predisposing to bacterial superinfections in influenza are related to altered "innate-like" unconventional T cells antibacterial activity but not to free fatty acids deficiency. The last mechanism is attributed to influenza but not to post-influenza bacterial superinfections.
Answer : The mechanisms leading to secondary bacterial infections post-influenza are extremely complex. Both “innate-like" unconventional T cells, such as NKT cells (Paget & Trottein (2019). Front Immunol. 10:336. doi: 10.3389/fimmu.2019.00336 ) and SCFAs, at least acetate (Sencio et al (2020). Cell Rep. 2020 Mar 3;30(9):2934-2947.e6. doi: 10.1016/j.celrep.2020.02.013), are involved. The second reference is quoted in the manuscript and justifies the use of the acetate-producer.
The most important failure is a total lack of the appropriate controls in the animal experiments demonstrating that the effect attributed to B.faecis supplementation is not just accidental and that no other bacteria used as a control not shown a similar effect. This is even more important in the situation in which the effects of the B.faecis supplementation were measured by death rates and weight loss only.
Answer: We fully agree with this comment and have now added a control (Supplementary Figure 4). We had administrated the F2 strain that is related to Coprococcus comes strain (described in table 1). With this strain, we did not observe any statistically significant protective effect in double-infected mice with S. pneumonia. These data are described in the results (yellow text) and in a new supplemental figure 4.
There are very recent reports in medical literature on bacterial pneumonia in ventilated patients in relation to Covid-19 infection and other virus infections. Moreover, the authors ideas that oral application of the B.faecis will be effective as prophylaxis of the bacterial superinfections in influenza are not sufficiently supported. The medical standards of the prophylaxis and treatment of influenza are stable and universal since many years, and they are based on highly effective vaccines and antiviral drugs. The same is true for pneumococcal pneumonia which is prevented by polysaccharide vaccines and treated with properly selected antibiotics. Why and what for an approach based on dietary supplementation with a bacterium producing acetate as the most important mechanism should be developed, commertialized and used in these infections?
Answer: We agree with this comment. We believe that some probiotics may reinforce our defenses against infections. It does not mean, that existing treatments are not efficient, it is only a suggestion to enrich our diet in at risk moment.
Minor remark: The authors should not use term “treatment” which should be secured for medicinal products only.
Answer : You are right, it should be better to use terms more adapted to nutritional strategies. So, we have replaced the terms “treatment “ and ” treated” by “administration”, “gavage” or “administrated” . See all along the text yellow-tracked words.
The figures 4; 5 and supplemental figure 3 have also been modified with “probiotic” instead of “treatment”.
Note: As recently requested by our administration, we have also affiliated some co-authors to a supplementary address: Paris Center for Microbiome Medicine (PaCeMM) FHU, AP-HP, Paris, Île-de-France, France.
Note : we have corrected an error in the legend of figure 4 and figure 5 (see in yellow).
Round 2
Reviewer 2 Report
There are still some minor errors to be corrected.
Author Response
All typography errors have been corrected